# Power of a randomization test in a single case multiple baseline AB design

Samantha Bouwmeester *, Joran Jongerling

Department of Psychology, Education and Child Studies, Erasmus University Rotterdam, Rotterdam, The Netherlands

* s.bouwmeester@essb.eur.nl

**Data Availability Statement:** All relevant data are within the paper and its Supporting Information files.

## Abstract

A randomization test can be used to statistically test hypotheses in multiple baseline designs to complement the commonly used visual inspection analysis. A crossed factor simulation study was performed to investigate the power of a randomization test in an multiple baseline design. The results show that the degree of autocorrelation of the observations, the number of participants, the effect size, the overlap of possible start moments of the intervention between participants, the ratio of the number of measurements in the baseline- and intervention phase, a gradually emerging effect, and the number of measurements had strong main effects on the power. The two-way interactions between number of participants and effect size, and between the number of measurements and the number of start moments of the intervention also had a large effect. An online tool was developed to calculate the power of a multiple baseline design given several design characteristics.

## Introduction

The single-case design has a long history in psychology as it was already used by famous founders like [1–4]. It is not restricted to the field of psychology, however, and can be used to inform and develop theory, evaluate the effectiveness of interventions, and study the behavior of organisms [5]. Although the kind of research questions involved in single case designs often differ from multiple case designs, single case designs may be viable alternatives for ordinary randomized trial designs when the number of participants is small, normality and homogeneity of variance assumptions are not warranted, or the sample is not random [6]. They are used in a clinical setting to evaluate the effect of a certain intervention on a small group of patients [7–9], but also in an educational context to test whether a manipulation can help students [10–12].

Although single case designs may vary greatly in their specific design properties, what is typical for all single case designs is that for each case, the outcome variable is measured repeatedly in each of two or more treatment conditions or phases (e.g., a baseline phase and an intervention phase). The effect of the intervention is evaluated by comparing the pattern of observed outcomes under the different treatment conditions, in which each case serves as its own control [13].

**Funding:** The author(s) received no specific funding for this work.

**Competing interests:** The authors have declared that no competing interests exist.

In this study we focus on the power of the multiple baseline across subjects AB design, abbreviated as across subjects MBD [14]. The across subjects MBD is the most common form of single case designs [15]. This across subjects design relies on group averages. [6] showed that a randomization test can also be used for single case designs with a single subject. However, the power of a single case single subject design is close to zero unless the effect of the intervention was huge and the number of measurements large (i.e. Cohen's d >1.5 and more than 40 measurements).

In the across subjects MBD design several measurements of some kind of outcome variable are administered to two or more subjects. These subjects are usually people, and we will therefore use the word participants throughout this manuscript but the MBD is applicable settings, behaviors, or groups as well. In the AB design all baseline measurements, A, precede the intervention measurements, B. This AB design differs from the alternation design in which baseline and treatment phases alternate. Although the AB design may have less internal validity than the alternation design which can better obviate history and maturation biases [14], the AB design has equal validity in situations where experimental control is demonstrated and it is often the only possible design for practical or ethical reasons. For example, in a clinical context, where the effect of some drug is evaluated one can mostly not alternate the phases where the drug is absent and where it is present. The AB design fits best in this situation.

In a multiple baseline AB design, the baseline phase ideally starts at the same time for each participant, while the intervention phase ideally begins at a different time for each participant. This is because start the baseline measurements at the same time but fluctuating the start time of the intervention for participants helps to guard against some threats to internal validity due to maturation or common history. Intervention phase patterns that are similar across participants are interpreted as evidence that the outcome responds to intervention [16]. This ideal design is often not applied in practice, however, which can be considered a methodological flaw [17].

Traditionally, the effect of an intervention in single case designs is evaluated using visual inspection of the pattern of observations. Visual inspection analysis offers a wide range of possibilities to investigate the patterns of individual time series and several measures have been developed to quantify this visual inspection (e.g. [18–21]). One may compare the means or medians of the observations in the baseline and intervention phase, or compare the range or standard deviation in the two phases. Alternatively, researchers can look at trend lines or inspect the percentage of non-overlapping observations (see https://architecta.shinyapps.io/SingleCaseDesigns/ for a tool to do the visual inspection analysis). Despite the obvious advantages and the intuitive attractiveness of visual inspection in single case designs, it has been criticized for high error rates and subjectivity [22–23]. [24] compared visual inspection with statistical analysis and concluded that the conclusions from visual analysis and statistical analysis had low level of agreement (see also [25–27]). Although the results of these studies are informative in that they confirm that statistical inference cannot be replaced by conclusions formed purely on visual inspection, we argue that the comparison of the two kinds of analyses which are inherently different is questionable. As the term already indicates, visual inspection is a meant to inspect the time series patterns and the effect of an intervention for one or a small number of participants. Statistical inference, instead, is aimed at generalization of a test statistic to some kind of population. Some researchers [28–29] even argue that because visual inspection analyses can merely detect major effects, these analyses lead to less Type I errors and an increase in Type II errors. We agree with [30] who recommended complementing visual analysis with a statistical analysis of the data, whenever possible.

When a researcher's goal is to statistically evaluate the mean effect of an intervention in a single case multiple baseline design, (s)he can't really use parametric tests, like *F* and *t* tests,

because data from single case designs violate the assumptions on which the parametric tests depend. That is, normality and homogeneity of variances can often not be guaranteed because the number of participants is small in single case designs. Moreover, the assumption of independent observations is problematic because data from single case designs are dependent, which can lead to autocorrelated residuals that can seriously bias the results from the parametric tests (see, e.g., [31–34, 23]). [14] explain that time series analysis can be used to handle the autocorrelated residuals, but this analysis method requires a lot more measurements than are usually available in single case MBD to detect the pattern of autocorrelations and identify the model [35–37].

A researcher could consider the nonparametric randomization test to analyze single case multiple baseline designs, since this test does not rely on any distributional assumptions (See S1 File, for some history of the randomization test).

In a multiple baseline design across subjects the randomization method may be based on the random assignment of participants to baselines. This method was presented by [38]. It may also be based on the random assignment of the start of the intervention for each of the participants [39]. [40] elaborated a combination of these two methods both randomizing the assignment of the participants to baselines and the start of the intervention. [41] compared this randomization test with those of [38] and [39], and concluded that the power is similar. Koehler and Levin's randomization test allows a more practical design because of the researcher defined staggered start moments of the intervention. In our study we therefore focused on this randomization procedure. We shortly explain the rationale of the Koehler and Levin's randomization here, for a complete example we refer the reader to the S2 File. Let $N$ be the number of participants under study, then the randomization procedure by Koehler and Levin's requires specifying $N$ separate ranges of start points for the intervention. If there are $N = 3$ participants that are measured 15 times each for example, these three ranges could be [*T5-T6*], [*T7-T8*] and [*T9-T10*] respectively, with $T$ representing time points (with *T5* representing the fifth time point). We won't go into the how these ranges should be determined in practice as it is beyond the scope of this article, but we do want to mention that in practice determining these ranges is often a difficult step because it depends on the minimum number of baseline- and intervention measurements that are required to get a stable baseline- and intervention estimate, where the minimum number of required measurements depends on the specific context. When the $N$ separate start ranges are determined, each consisting of $k$ possible start moments, all possible combinations of participant $i$ ($i = 1,...,N$) and start point $k$ are determined leading to $N! \prod_{i=1}^{i=N} k_i$ permutations. Note that the total number of permutations is smaller when there is an overlap in the range of possible start moments of the intervention for different participants [14]. One of these permutations, that is, one of the combinations of participants and start moment of the intervention is used in the actual data collection, and the mean baseline- and intervention scores are calculated based on the specific start moments used. Next, baseline- and intervention scores are calculated form the observed data, using each of $N! \prod_{i=1}^{i=N} k_i$ permutations of individual and start moments. The mean differences averaged over all participants of all permutations together form the distribution of the randomization test. Note that this distribution does not rely on any distributional assumptions, and is not likely to be symmetric. Finally the $p$-value is calculated by dividing the number permutations that have an averaged mean difference equal or more extreme than the observed averaged mean difference by the total number of permutations. The power of the randomization test is defined as the probability that the null hypothesis is correctly rejected.

There are two things important to realize. First, because the shape of the distribution is unknown, the randomization test is one-tailed, instead of two-tailed. Second, the null

hypothesis of this randomization test is *not* that the mean baseline is equal to the mean intervention score. Because there is a minimum number of baseline and a minimum number of intervention observations which are not part of the randomization, the mean difference between the baseline scores and the intervention scores may not be zero. Instead, the null hypothesis is that the mean difference between baseline and interventions observations is equal for all possible permutations. From this it follows that the Type I error rate is the probability that one decides that the mean difference between baseline and interventions observations is *not* equal for all possible permutations when there is in fact no effect for permutations. The Type II error is the probability that one decides that the mean difference between baseline and interventions observations is equal for all possible permutations when there is an effect for permutations.

Although the randomization test does not rely on distributional assumptions, there is a necessary and sufficient condition when using a randomization distribution to obtain valid statistical significance [42–43]. This is the exchangeability assumption which states that observations can be exchanged with other observations without loss of meaning to the grouping/sequence. In a single case design multiple measurements within a relatively short interval are taken from the same person and these observations will nearly always be autocorrelated to at least some degree [44]. This autocorrelation on the measurement level will, although to a less extent, be reflected in the test statistic of the randomization test which violates the exchangeability condition.

Some researchers argue that autocorrelated data do not affect the statistical validity of randomization tests if the amount of data per phase is sufficiently large [14, 45, 44]. [38] and [46] have suggested, respectively, that autocorrelation equally affects all the permuted data in the randomization distribution and that randomization tests overcome autocorrelation problems. However, [47] (see also [48–49, 43, 50–51]) are critical about the validity of the randomization test when observations are autocorrelated. [41] are, as far as we know, the only researchers who evaluated the effect of autocorrelation in an AB across subjects MBD. They concluded that it is important to take the level of autocorrelation into account when investigating the power of MBD's.

They showed that the Koehler-Levin randomization test can control Type I error rates even with a considerable amount of autocorrelation, the power of the randomization test, however, was negatively related to the autocorrelation.

Besides the autocorrelation there are several other factors which may influence the power of the Koehler Levin randomization test in a single case AB multiple baseline design. A few of these factors, like the number of participants and the number start moments of the intervention and the number of measurements in the baseline- and intervention phases may be controlled by the researcher. Others, like the effect size, whether there is a correlation between the mean of the baseline and the mean of the intervention scores and whether the variation in the outcome is similar in the two phases will mostly be determined and restricted by the context of the research, and the researcher can mostly not change these factors to enhance power. The goal of this paper is to investigate the effect of these factors on the power in a range of practically realistic scenarios. We think that researchers using a randomization tests to evaluate their MBD may really be interested in the results of this study because there is hardly any literature on this topic nor is there software available which can be used to a priori evaluate the power of a particular design.

Before we will describe the details of the simulation study that was performed to evaluate the effect of the factors we will first explain the factors in a bit more detail.

First, as explained above, the level of autocorrelation between the observations within a participants is expected to effect the power. Just like [41] we took a range of autocorrelations into

account from 0 to .5. The higher the autocorrelation, the lower the expected power. We had no reasons to expect this effect to interact with other factors of the design (see S4 File).

Second, an across subjects MBD requires at least two participants. The larger the number of participants the larger the number of permutations. A minimum number of participants is required to be able to reject the null hypothesis, since the *p*-value is calculated by dividing by the total number of permutations. [41] showed, though, that the total number of permutations may not be related to the power of the design.

Third, the larger the number of possible start moments of the intervention per participants, the larger the number of permutations. The actual range of possible start moments may be delimited by the number of participants, the minimum number of observations in each phase and the total number total number of measurements. In many contexts quite some baseline observations are required to get a stable estimate of the baseline score of an individual. In this case the range of possible start moments of the intervention may not be large. Note that, as a consequence, the number of permutations for the randomization test may be too small to reach statistical significance.

Fourth, in a multiple baseline design, the number of measurements may differ for participants. In general, the more measurements, the more reliable the test statistic because it is based on more observations. It depends on the actual context in which the outcome is measured how much observations are required to get stable estimates of the baseline and the intervention scores.

Fifth, the within participant effect size will have an influence on the power. Multiple kinds of effect sizes for single case designs have been discussed [8, 52]. In our study we defined the within participant effect size as the mean difference of a participant's baseline and intervention scores divided by the pooled standard deviation of the scores in the baseline and the intervention phases. [41] found that a Cohen's *d* of at least 1.5 is required to have sufficient power (power = .80). In their study they investigated the effect of effect sizes .5, 1, 1.5 and 2 in a design with four participants and two start moments of the intervention. The power of *d*'s of .5 and 1 was very low for all randomizations tests they compared.

Sixth, in order to get stable mean estimates for both the baseline and the intervention phase it seems preferable to have as many measurements as possible in both phases. However, this may not be preferred from a practical point of view. In almost all clinical contexts where the intervention is a treatment and the baseline observations are collected when someone is on the waiting list to be treated, one wants to start the treatment as soon as possible. Moreover, more observations during the treatment will often be preferred over an equal number of observations in the baseline and the intervention. The question is whether a smaller number of measurements in the baseline phase than in the intervention phase has a negative effect on the power, compared to an even number of observations in both phases.

Seventh, the power may be influenced by an overlap in the range of possible start moments of the intervention for different participants. Although unique possible start moments are preferred, the actual context may not allow a nonoverlapping ranges. This may be the case when the number of measurements is small or when a large number of baseline measurements is required to get a stable baseline score. An overlap in possible start moments in combination with a small number of participants leads to smaller number of permutations which may have a negative effect on the power.

Eighth, in clinical contexts, the effect of an intervention may often be correlated with the mean baseline scores across subjects. Note that this effect can exist apart from, or in addition to, the autocorrelation of the observations within a participant. We investigated whether correlated baseline and intervention means across subjects have a negative effect on the power.

Ninth, the ratio of variation in scores in baseline and intervention phase may have an influence on the power. In some contexts one may expect a homogenous variation in scores in the baseline and the intervention phase. In many contexts however, the intervention will cause the observations to become more similar or just more variable. While the mood of depressive people may be stably low when on the waiting list, it may become more variable because of the effect of the psychotherapy treatment. Contrary, people suffering from a bipolar disease may become less alternating in mood as soon as they got the right medicine. We investigated what the effect is on the power of this heterogeneity in variances between the scores in the baseline and the intervention phase.

Finally, the effect of an intervention may appear suddenly, directly after the onset of the treatment or emerge gradually during the intervention. The effect-size of a gradually emerging effect is obviously smaller than of a suddenly appearing effect (see [53] for an overview of effect sizes with gradually emerging effects). In our study we evaluated the effect of a gradually emerging effect on the power of the randomization test.

The different factors may not only have main effects on power, but could also possibly interact. In our simulation study we therefore used a crossed design of all factor levels, except for the factors autocorrelation and gradually emerging effect which we didn't expect to interact. Because higher order effects may be less informative in general, we only discuss the results of the main- and two way interactions. Although the general results on the main- and two way interaction effects of the factors on the power presented below contribute useful knowledge to the relatively sparse literature about randomization tests in MBD's, their usefulness may be limited for researchers who wants to know whether their *specific* MBD with several interacting factors has sufficient power. In order to enable researchers to study the power of their specific design we developed an online tool (https://architecta.shinyapps.io/power_MBD/). This tool can be used to evaluate the higher order interaction effects, gives a power estimate for the specific design, and shows how changes in the design properties influence the power. Moreover, we offer researchers the opportunity to do their own simulation study in which they can simulate the power of their own MBD and which is not restricted to the levels of the factors we included in our simulation study.

## Method

The manuscript and supporting information sections can be viewed at http://dx.doi.org/10.17504/protocols.io.9vrh656.

### Simulated Factors

In order to evaluate the effect of the above-mentioned factors on the power of the randomization test in a AB across subject MBD a simulation study was performed. Table 1 shows the factors and their levels. The autocorrelation between the observations within a participant were 0, .1, .2, .3, .4 and .5 following [41]. The number of participants was varied from 2 to 12. The number of possible start moments per participant varied from 2 through 4. The number of total measurements was 15, 30 and 60. For the effect size Cohen's $d$ was used having the values .3, .6 and 1. The effect size was simulated for each participant separately. The standard deviation within the baseline and intervention phase could be identical (both 1); the standard deviation in the baseline phase could be twice as large as the standard deviation in the intervention phase ($sd_B = 1.33$, $sd_I = .67$); or the standard deviation in the intervention phase could be twice as large as the standard deviation in the baseline phase $sd_B = .67$, $sd_I = 1.33$); For each participant baseline scores were simulated for all measurements $t$ by the autoregressive function $B_t = AR^*B_{t-1} + e_t$, where $AR$ varied from 0 to 0.5, and $e$, the error, is drawn from a normal

**Table 1. Factors and their levels of the simulation study.**

| Factor | Levels |
|---|---|
| Autocorrelation (*AR*) | *AR* = 0, 0.1, 0.2, 0.3, 0.4, 0.5 |
| Number of participants (*i*) | *i* = 2, .., 12 |
| Number of possible start moments intervention (*k*) | *k* = 2, 3, 4 |
| Number of measurements (*t*) | *t* = 15, 30, 60 |
| Ratio *sd* in baseline (B) and intervention (I) | $sd_B = sd_I$; $sd_B = 2sd_I$; $2sd_B = sd_I$ |
| Effect size (*d*) | *d* = .3, .6., 1 |
| Equal # of scores baseline & intervention | TRUE; FALSE |
| Non-overlap of possible start moments | TRUE; FALSE |
| Correlated mean baseline and mean intervention | TRUE; FALSE |
| Gradually emerging effect (portion of intervention measurements) | 0, ¼, ⅓, ½ |

distribution with mean = 0 and standard deviation equal to the standard deviation of the baseline scores (i.e. .67, 1, or 1.33). Note that $B_{t = 1} = e_{t = 1}$. The autoregressive function for the scores in the intervention is the same, $I_t = AR^*I_{t-1} + e_t$, except that the error is drawn from a normal distribution with mean of .3, .6, or 1, and standard deviation equal to .67, 1, or 1.33.

Furthermore, the number of baseline and intervention phase measurement was similar or there were more observations in the intervention phase than in the baseline phase. The number of observations was at least two within each phase. The start moments per participant could be unique or overlap. When the start moments were unique all participants had different possible start moments. For example, when there were three participants, the number of start moments was four, and the total number of measurements was 30, the range of possible start moments could be 4, 5, 6, 7; 8, 9, 10, 11; and 12, 13, 14, 15 while in the overlapping situation the range of possible start moments could be 4, 5, 6, 7; 5, 6, 7, 8; 6, 7, 8, 9. Note that large overlap in start moments was chosen to capitalize on the effect of overlap.

The mean score in the baseline phase was related to the scores in the intervention phase or not. When the means were correlated the autoregressive function to simulate the intervention scores was $I_t = \bar{B} + AR * I_{t-1} + e_t$. Note that this equals a correlation of .7 between the baseline and the intervention means.

When all factor levels—except for the factors autocorrelation and gradually emerging effects which were both fixed to 0 (See S4 and S5 Files, for an explanation)—were crossed there were 648 different scenario's. Some combinations of factor levels were impossible, however. This was the case when the number of measurements was 15 or 30. S1 Table shows which combinations were impossible. Moreover, a minimum number of permutations is required make statistical testing meaningful. That is, with less than 20 permutations, the *p* value cannot become smaller than a Type I error rate of .05. This is the case when the number of participants is 2 and the number of possible start moments is 2 or 3. The power is per definition 0 in these situations and the randomization test should not be considered.

In order to study the effect of autocorrelation of the observations within a participant on the power of the MBD we did a separate simulation study in which we evaluated the effect of autocorrelation for a default situation and nine alternative situations which each differed from the default situation with respect to one factor level. These ten situations were simulated for four and eight participants. Table 2 shows the factor levels of the default situation and the alternative situations.

In order to study the effect of a gradually emerging effect on the power of the MBD we also did a separate simulation study in which we again evaluated the effect of a gradually emerging effect for a default situation and nine alternative situations which each differed from the

**Table 2. Overview of the default and nine alternative designs used to investigate the effect of autocorrelation on power.**

| Situation | # of possible start moments, $k$ | correlated B[1] and I[2] Mean, $r_{BI}$ | Equal # of obs. in B and I, $\#obs_B = \#obs_I$ | Non-overlapping $k$ | number of measurements, $t$ | Mean diff. B and I, $d$ | $sd$ B | $sd$ I |
|---|---|---|---|---|---|---|---|---|
| default | 3 | FALSE | TRUE | TRUE | 60 | 1 | 1 | 1 |
| $k = 2$ | 2 | FALSE | TRUE | TRUE | 60 | 1 | 1 | 1 |
| $k = 4$ | 4 | FALSE | TRUE | TRUE | 60 | 1 | 1 | 1 |
| $r_{BI}$ | 3 | TRUE | TRUE | TRUE | 60 | 1 | 1 | 1 |
| $\#obs_B = \#obs_I$ | 3 | FALSE | FALSE | TRUE | 60 | 1 | 1 | 1 |
| Non Overlap $k$ | 3 | FALSE | TRUE | FALSE | 60 | 1 | 1 | 1 |
| $d = .3$ | 3 | FALSE | TRUE | TRUE | 60 | 0.3 | 1 | 1 |
| $d = .6$ | 3 | FALSE | TRUE | TRUE | 60 | 0.6 | 1 | 1 |
| $sd_B = 2sd_I$ | 3 | FALSE | TRUE | TRUE | 60 | 1 | 1.33 | 0.67 |
| $2sd_B = sd_I$ | 3 | FALSE | TRUE | TRUE | 60 | 1 | 0.67 | 1.33 |

Shaded cells show the factor level that is different from the default scenario.

[1]B = Baseline,

[2]I = Intervention

default situation with respect to one factor level. These ten situations were simulated for six participants. Table 3 shows the factor levels of the default situation and the alternative situations. Next we formulated ten alternative designs in which only one of the factor levels was different from the default design. For these ten designs we simulated data in the same way as was done for the simulations in the main text with a suddenly emerging effect and gradually emerging effects during ¼, ⅓, and ¼ of the treatment measurements.

## Simulation study

Program R [54] was used to run the simulations. See S3 File for the R code.

**Table 3. Overview of the default and nine alternative designs used to investigate the effect of gradually emerging effects on power.**

| Situation | # of possible start moments, $k$ | Equal # of obs. in B[1] and I[2], $\#obs_B = \#obs_I$ | AR | number of measurements, $t$ | Mean diff. B and I, $d$ | $sd$ B | $sd$ I |
|---|---|---|---|---|---|---|---|
| default | 3 | TRUE | 0 | 60 | 1 | 1 | 1 |
| $k = 2$ | 2 | TRUE | 0 | 60 | 1 | 1 | 1 |
| $k = 4$ | 4 | TRUE | 0 | 60 | 1 | 1 | 1 |
| AR = .5 | 3 | TRUE | .5 | 60 | 1 | 1 | 1 |
| Nr. Measurements = 30 | 3 | TRUE | 0 | 30 | 1 | 1 | 1 |
| $\#obs_B = \#obs_I$ | 3 | FALSE | 0 | 60 | 1 | 1 | 1 |
| $d = .3$ | 3 | TRUE | 0 | 60 | 0.3 | 1 | 1 |
| $d = .6$ | 3 | TRUE | 0 | 60 | 0.6 | 1 | 1 |
| $sd_B = 2sd_I$ | 3 | TRUE | 0 | 60 | 1 | 1.33 | 0.67 |
| $2sd_B = sd_I$ | 3 | TRUE | 0 | 60 | 1 | 0.67 | 1.33 |

Shaded cells show the factor level that is different from the default scenario.

[1]B = Baseline,

[2]I = Intervention

In the randomization test the *p*-value was calculated by dividing the number of permutations that have a test statistic as extreme or more extreme than the observed test statistic by the total number of permutations. Because the number of permutations becomes very large with an increasing number of participants we draw a sample of 400 permutations from the total number. A pilot study showed that the *p*-value of this sample of 400 permutations was very similar to the results of the complete permutation distribution. In order to calculate the power, 500 replications were simulated and the power was calculated by dividing the number of replications with a *p*-value below .05 (Type I error rate) by 500. A pilot study showed that 500 replications were sufficient to get a stable estimate of the power. For each of the scenario's the power was calculated 100 times. A pilot study showed that 100 replications were sufficient to get a stable estimate of average power.

## Results

### Type I error

Before we evaluated the power we first checked the type one error rates for all factor levels. The nominal Type I error rate was .05. S2 Table shows the actual mean Type I error rates and the standard deviation. Because not all factor levels could be crossed when the number of measurements were equal to 15 or 30 (see Table 2) the Type I error rates were evaluated for the scenario's having 60 measurements. The Type I error rate was .01 when there were two participants. This means that the power for a design with only two participants will be low because of the low error rate. For three, four and five participants the actual error rate was a bit lower than the nominal error rate (resp. .046, .049, .048). From six participants the Type I error rate is .05. Based on these results we decided to only include six or more participants to evaluate the Type I error rate for the other factors. For all factors except the ratio of the standard deviation in the baseline and the intervention phase the mean Type I error rate was .05. When the standard deviation in the baseline phase was half the standard deviation in the intervention phase the Type I error rate was .051, slightly liberal. When the standard deviation was larger in the baseline phase or when the standard deviations were equal the Type I error rate was .049, slightly lower than the nominal Type I error rate.

In order to investigate the effect of the number of measurements on the Type I error rate we took a subsample with only eight participants and overlap in start moments of the intervention. With this selection all other factors levels could be crossed. When the number of measurements was 15, the Type I error rate was .049. With 30 and 60 measurements the Type I error rate was .05.

### Power

**Effect of autocorrelation on power.**   The power was calculated for the ten situations described in Table 3 for four and eight participants and autocorrelations 0, .1, .2, .3, .4 and .5. Fig 1 shows the results for four and eight participants. For all of the 20 situations, there is negative relationship between the autocorrelation and the power. The effect of the autocorrelation on the power doesn't seem to interact with other design features. In S4 File, we showed that the power for a specific design with autocorrelated data could almost perfectly be predicted by the power and the standard deviation of the power of that design when the data were not autocorrelated.

**Effect of a gradually emerging effect on power.**   Table 4 shows the power and effect size of the ten scenario's in four situations. In the first situation the effect is suddenly emerging from the first treatment measurement. In the other situations the effect was emerging during respectively ¼, ⅓, and ½ of the treatment measurements. The results show that the effect of a

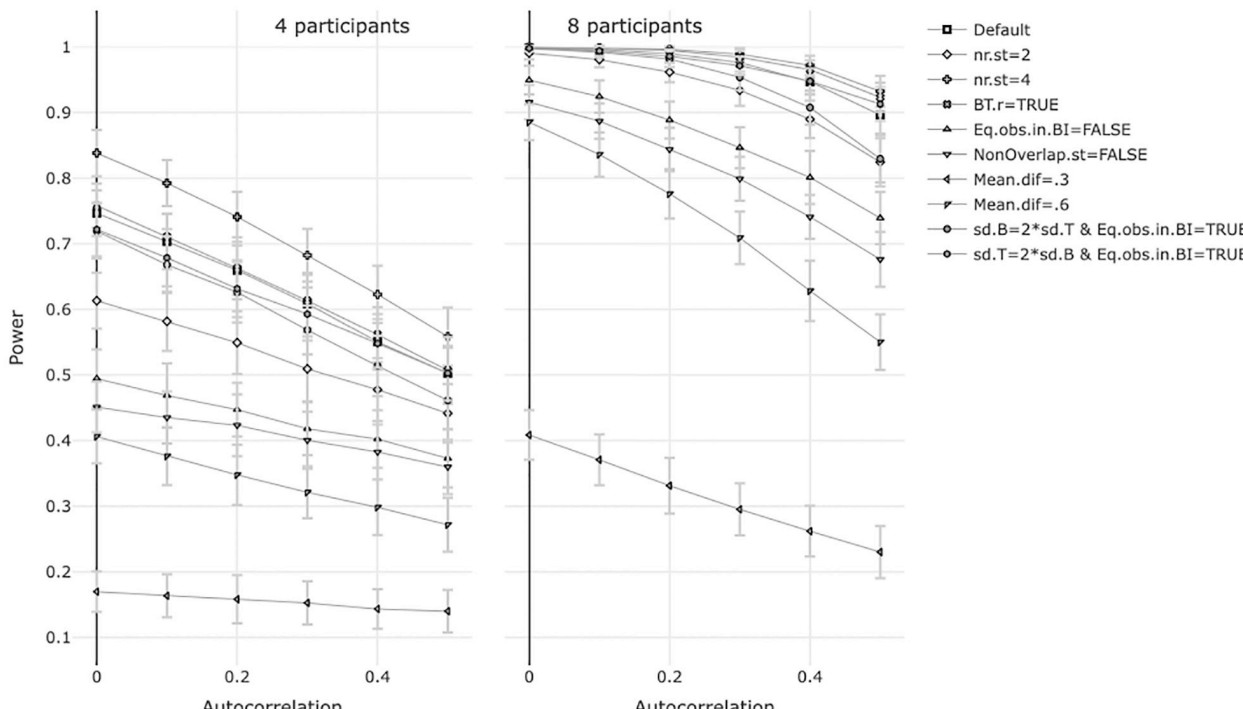

**Fig 1. Effect of autocorrelated observations on power for 10 designs with 4 and 8 participants.**

gradually emerging effect on the power is strong and negative. For all scenario's the power drops much more than the effect size does.

Most of the times researchers will be interested in the final effect of the intervention and not in the average effect in which the gradually emerging effect measurements are included. In S5 File, we evaluated the power of a randomization test when corrected for the gradually emerging effect.

**Table 4. Power and effect size for the default and nine alternative designs used to investigate the effect of gradually emerging effects.**

| | Portion of intervention measurements at which the effect is emerging | | | | | | | |
|---|---|---|---|---|---|---|---|---|
| | **0** | | **¼** | | **⅓** | | **½** | |
| | *Power* | *ES*[*] | *Power* | *ES* | *Power* | *ES* | *Power* | *ES* |
| default | .97 | 1.00 | .40 | 1.01 | .26 | 0.89 | .15 | 0.77 |
| $k = 2$ | .91 | 1.00 | .21 | 1.01 | .15 | 0.89 | .10 | 0.77 |
| $k = 4$ | .99 | 1.00 | .58 | 1.01 | .39 | 0.89 | .21 | 0.77 |
| Autocorrelation = .5 | .80 | 1.00 | .25 | 1.01 | .18 | 0.89 | .09 | 0.76 |
| Nr. Measurements = 30 | .74 | 1.00 | .17 | 1.00 | .13 | 0.88 | .12 | 0.77 |
| #obs$_B$ = #obs$_I$ | .94 | 1.00 | .70 | 1.02 | .48 | 0.90 | .26 | 0.78 |
| $d = .3$ | .29 | 0.30 | .12 | 0.30 | .09 | 0.27 | .07 | 0.23 |
| $d = .6$ | .70 | 0.60 | .22 | 0.61 | .16 | 0.53 | .10 | 0.46 |
| $sd_B = 2sd_I$ | .96 | 1.00 | .37 | 1.01 | .24 | 0.89 | .14 | 0.77 |
| $2sd_B = sd_I$ | .96 | 1.00 | .38 | 1.01 | .26 | 0.89 | .07 | 0.65 |

[*]ES = effect-size

**Effect of other design factors on power.** Since the effect of autocorrelation on the power could well be predicted by the power and standard deviation of the power when the autocorrelation was 0, we evaluated the effect of the factors on the power for data in which the autocorrelation was 0. The online tool (https://architecta.shinyapps.io/power_MBD/) can be used to calculate the power for a given design and a range of autocorrelations ($AR = 0 - .5$). An analysis of variance was performed to evaluate the effect of the other factors on the power of the randomization test. Because not all factor levels could be crossed when the number of measurements was equal to 15 or 30 (see Table 2) the first analysis was performed on the scenario's having 60 measurements. We included all main- and two-way interaction effects. Although higher order interaction effects could be evaluated, we think, a general interpretation of these effects is less informative. For calculating the effect of a specific AB across subjects MBD on the power we refer the reader to the online tool which can be used to graphically evaluate the effect of higher order interaction effect as well as calculating the power for a specific combination of factor levels. Since the number of simulated power estimates were large (100) in each cell of the design the significance of the main and interaction effects is not very informative. We therefore focused on the effect size partial eta squared, $\eta_p^2$. Effect sizes of .02 were interpreted as small, .13 as medium and .26 as large [55].

Fig 2 shows the main effects of the seven factors as well as the effect sizes, partial eta squared. Note that the power estimates of the individual factor levels are averaged over all other factors. For example, in panel A, the estimated power (.6) for eight participants is averaged over effect sizes $d = .3$, $d = .6$ and $d = 1$. This makes it difficult to interpret the trends in an absolute way. The effect size, $\eta_p^2$, of the number of participants (Panel A, Fig 2) is .95, which

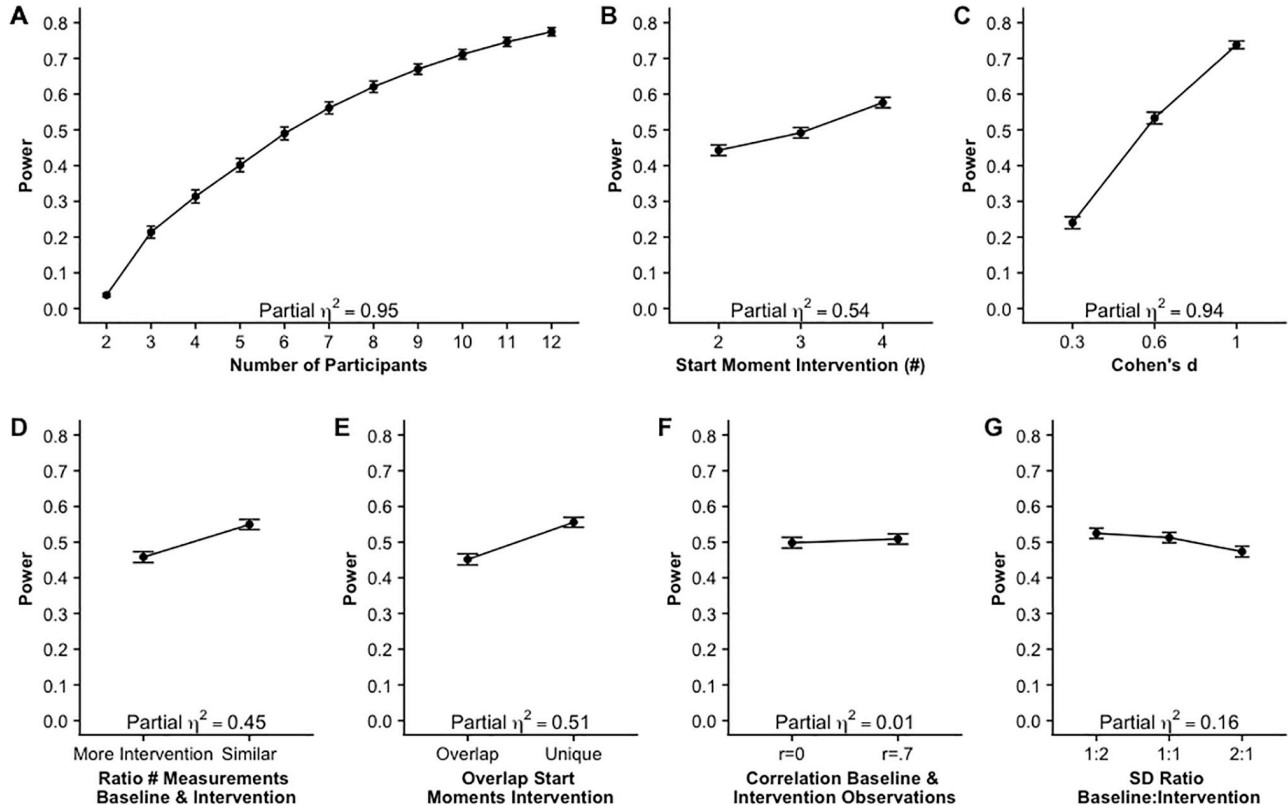

**Fig 2. Main effects and effect sizes of seven factors on power.**

is very large. The power increases fast with a larger number of participants. The effect of the effect size on the power (Panel C, Fig 2) is also very large ($\eta_p^2 = .95$). The number of start moments of the intervention (Panel B, Fig 2) has a large partial eta squared ($\eta_p^2 = .54$), and the power increases when the number of possible start moments increases from 2 to 3 start moments and the effect on power is even stronger from 3 to 4 start moments. When the number of measurements is similar in the baseline and the intervention phase the power is larger than when there are more measurements in the intervention phase (Panel D, Fig 2). This effect is large, $\eta_p^2 = .45$. Averaged over all other factors, unique possible start moments result in higher power than overlap in start moments. This effect is large as well, $\eta_p^2 = .51$. The effect of the ratio in standard deviation between the baseline and the intervention phase (panel F, Fig 2) is of medium size, $\eta_p^2 = .16$. The power is largest when there is more variation in the intervention phase than in the baseline phase and smallest when there is more variation in the baseline phase than in the intervention phase. The effect size of the effect for correlated baseline and intervention means is close to 0 (panel E, Fig 2), indicating that a correlation of .7 between the mean of the baseline and the mean of the intervention measurements does not result in a different power estimate than a correlation of 0.

Fig 3 shows all two-way interactions between number of participants and all other factors. The strongest interaction effect is the interaction between number of participants and effect size (Panel B, Fig 2), $\eta_p^2 = .73$. When the effect size Cohen's $d$ is .3 (small effect) the power will not exceed .5 even when there are twelve participants. Having a medium effect size ($d = .6$), eight participants are required to have sufficient power (.8). Having a large effect size ($d = 1$), sufficient power can be reached with only six participants. Note, again, that these power estimates were obtained by aggregating over all other factors. A medium effect, $\eta_p^2 = .11$, was found for the interaction between number of participants and overlap in start moments of the intervention. Panel D in Fig 3 shows that the benefit of non-overlapping possible start moments of the intervention is larger when there is a larger number of participants. A somewhat smaller interaction effect, $\eta_p^2 = .09$, was found between number of participants and the ratio number of measurements in the baseline and the intervention phase (Panel C, Fig 3). When there are two or three participants there is hardly a difference between a similar number

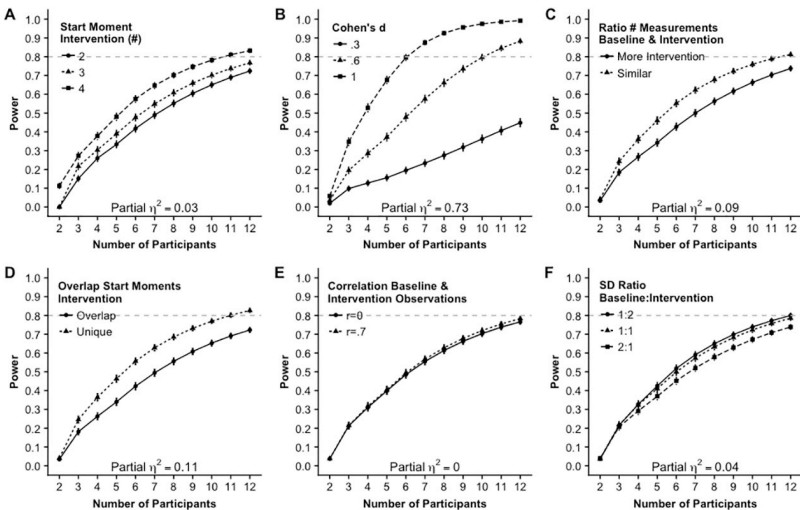

**Fig 3. Two way interaction effects on the power including number of participants.**

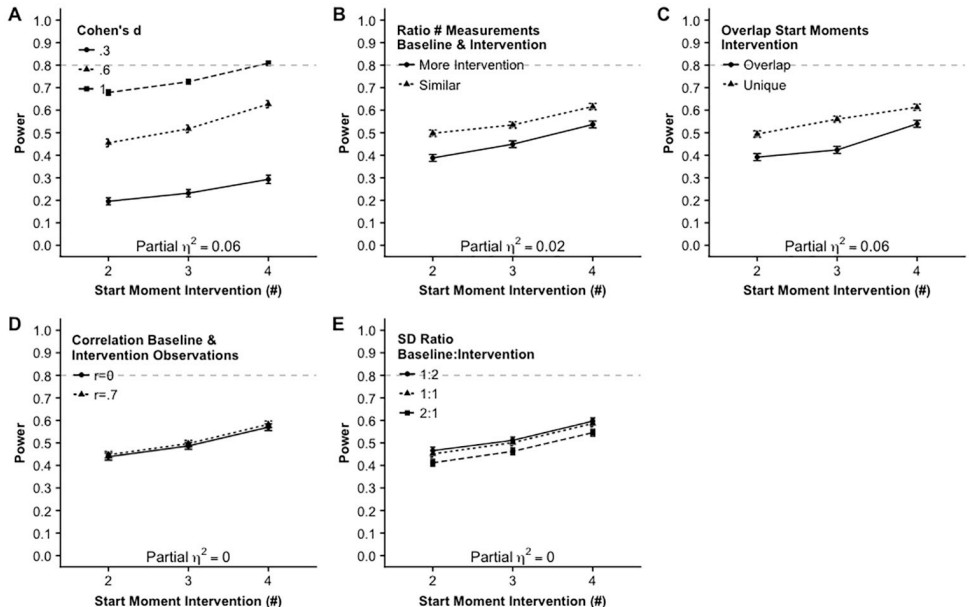

**Fig 4. Two way interaction effects on the power including possible start moments of the intervention.**

of measurements in both phases or more measurements for the intervention phase. From four participants there is benefit of a similar number of measurements in both phases. Fig 3, panels A, E and F show that the other two-way interaction effects with number of participants are small or absent.

Fig 4 shows that the remaining two-way interaction effects including start moment intervention are small or negligible. Although the main effect of effect size Cohen's *d* is huge, there are no interaction effects with the other factors except for number of participants. Fig 5 shows

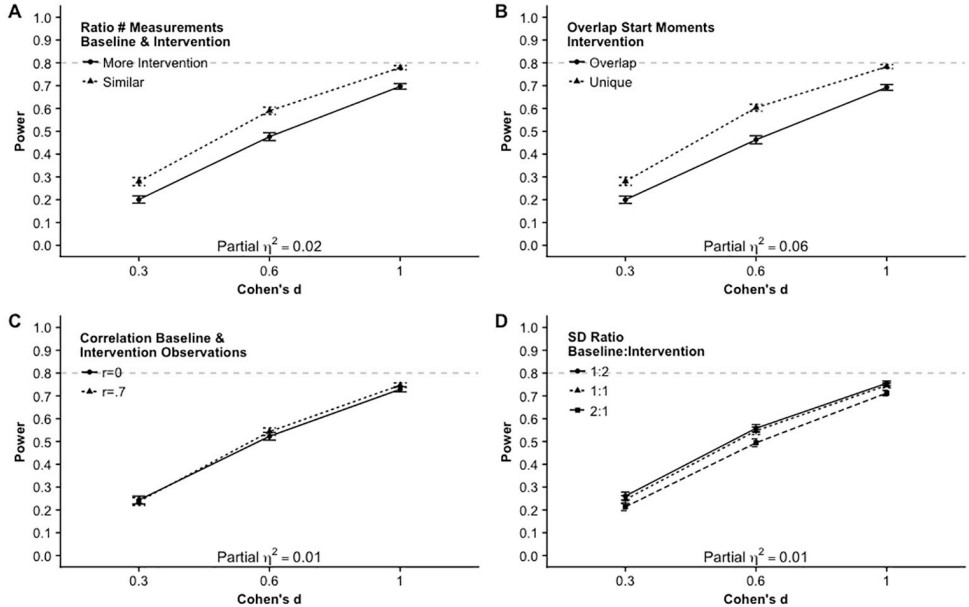

**Fig 5. Two way interaction effects on the power including effect size, Cohen's *d*.**

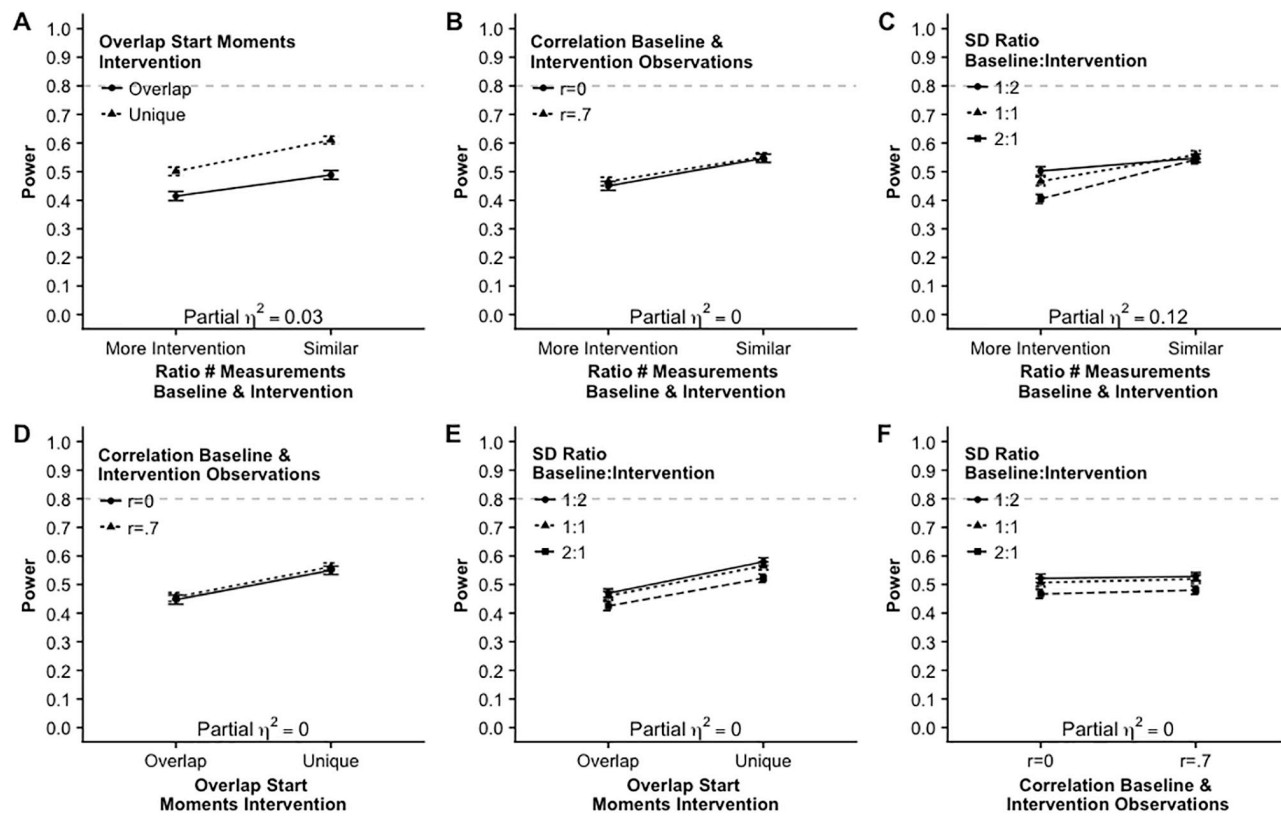

**Fig 6. Remaining two way interaction effects on the power.**

that the remaining two-way interaction effects including Cohen's *d* are very small. Fig 6 shows the remaining two way interaction effects. These effects are very small except for the medium effect size, $\eta_p^2 = .12$, for the interaction between the ratio of the number of baseline and intervention measurements and the ratio of the standard deviation within the baseline and the intervention measurements (Fig 6, Panel C). When there are more measurements during the intervention phase and the standard deviation is larger in the baseline phase the power is smaller than in all other combinations. It turned out that when there is a similar number of observations in the baseline and the intervention phase that the ratio of the standard deviation does not affect the power.

**Number of measurements.**   S1 Table showed that with 15 and 30 measurements not all factors could be crossed. In order to investigate the effect of the number of measurements and its interactions with the other factors we took a subsample with only eight participants and overlap in start moments of the intervention. With this selection all other factors levels could be crossed. We did an analysis of variance and again focused on the effect size to evaluate the effect of number of measurements and its two-way interactions. Note again that the power estimates for one level or a combination of levels was obtained by aggregating over the remaining factors.

Fig 7, Panel A shows that there is a large effect of number of measurements, $\eta_p^2 = .43$. The increase in power from 15 to 30 measurements is larger than the gain in power between 30 and 60 measurements. The interaction between number of measurements and start moments of the intervention is also large, $\eta_p^2 = .38$. Panel B, Fig 7 shows that for two or three possible start moments there is hardly any difference between the number of measurements levels but

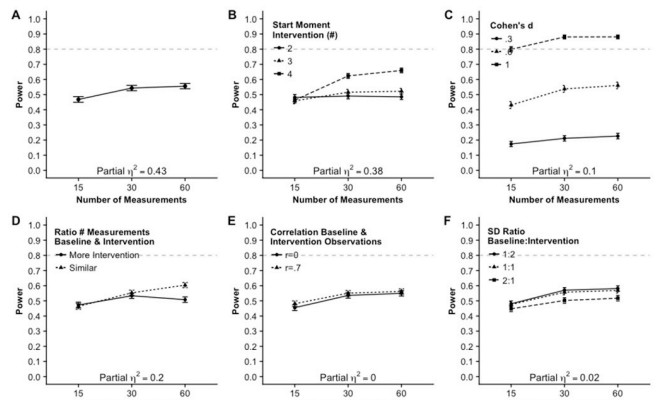

**Fig 7. Main- and two-way interaction effects including number of measurements on the power.** Note that number of participants is eight and possible start moments of the intervention overlap.

when there are four possible start moments of the intervention the power increases considerably from 15 to 30 measurements. The interaction effect of number of measurements and Cohen's $d$ (Panel C, Fig 7) turned out to be of medium size, $\eta_p^2 = .1$. With a small effect size ($d = .3$) the number of measurements hardly affected the power. When the effect is medium ($d = .6$) or large ($d = 1$) there is a considerable gain in power from 15 to 30 measurements.

Panel D in Fig 7 shows a medium to large effect size, $\eta_p^2 = .2$, for the interaction effect between the number of measurements and the ratio of the number of measurements in the baseline and the intervention phase. When there is a similar number of measurements in both phases, there in an increase in power from 15 to 30 and 30 to 60. When there are more measurements in the intervention phase, however, the gain in power from 30 to 60 measurements is not present.

## Discussion

In this study we provided information about the influence of several factors on the power of a randomization test in a single case multiple baseline across subjects design. The results showed that autocorrelation of the observations has a negative effect on the power. This effect did not interact with any other design properties. However, the effect of autocorrelation of the observations on the power turned out to be a function of the power and the standard deviation of the power when the autocorrelation was 0.

The number of participants had a large effect on the power as well as the within participant effect size. With small within participant effect sizes (Cohen's $d = .3$), the usefulness of a randomization test is limited as it has hardly any power even with twelve participants (power < .5). With a medium effect, Cohen's $d = .6$, sufficient power (.8) is reached from ten participants. Having a large effect, Cohen's $d = 1$, six participants already result in an expected power of .8. These results may seem disappointing at first glance, given that medium and large effect sizes are more rare than common in experimental designs in the social sciences (However, see [56] for an empirical evaluation of Cohen's effect size guidelines in the context of individual differences.). However, one may not compare the context in which these large $n$ randomized trials designs take place with the single case multiple baseline design contexts. Single case design have often been used in educational or clinical contexts in which one wants to evaluate an intervention or a therapy which effect has already been established or proven in the educational or clinical population from which the participants originate. The goal of the single case

studies is often not to evaluate whether the results of an intervention can be generalized to the population but rather to evaluate if an intervention is effective in a certain subgroup. The focus of single case studies is more on internal than on external validity [57]. In such situations medium or large effects are far more likely.

The number of possible start moments of the intervention also had a strong main effect on the power. Taking three instead of two possible start moments leads a significant increase in power and this increase is even larger from three to four possible start moments. This effect hardly interacts with any of the other factors taken into account. More possible start moments leads to a larger number of combinations in the permutation distributions. Following [41], we did not expect this larger permutation distribution to have a positive effect on the power beforehand. An explanation may be found in the way we chose the start moments in our design. We defined the possible start moments consecutively leading to a wider range of possible start moments over participants when there were more possible start moments per participant. Take for example a situation with three participants, two possible start moments and 60 measurements. This could lead to the following set of start moments [27, 28]; [29, 30]; [31, 32]. The first possible start moment of the intervention is the 27$^{th}$ measurement and the last possible start moment is the 32$^{nd}$ measurement. In the same situation but now with three possible start moments, the start moments could be [26, 27, 28]; [29, 30, 31]; [32, 33, 34], leading to the wider range from 26 through 34. As you can see, in there is a confounding effect when choosing more possible start moments, a wider range. It is probably not the number of possible start moments per participants, but the wider range of possible start moments over participants which leads to a higher power. One may, of course, solve this confounding problem by fixing the range and choosing non-consecutive possible start moments. We think, however, that non-consecutive possible start moments are exceptional and successive possible start moments are generally preferred in practice.

The same line of reasoning can be used to explain the effect of the non-overlap of possible start moments of the intervention. Non-overlapping possible start moments result in a higher power than overlapping start moments. In the overlap condition the range of possible start moments is smaller than in the unique possible start moments condition. This results in a confound. Choosing for overlapping start moments may not be an intended choice of the researcher but merely imposed by practical restrictions. In a clinical context, for example, it may be required that a therapy starts within a certain range of measurements.

For a higher power of the randomization test it is preferred to choose a similar number of baseline and intervention measurements. This effect is easily explained by the fact that more measurements lead to more stable estimates of the mean. We compared only two conditions here because the situation with more baseline than intervention measurements may not be feasible in practice. There were some remarkable interactions of this factor. One is the number of measurements. With only 15 measurements, the difference in number of measurements in the baseline and intervention phase between the two conditions (equal # observations vs. more intervention observations) is relatively small leading to no difference in power for the two conditions. With 60 measurements, however, the difference in number of measurements in the baseline and intervention phase between the two conditions is large, leading to a higher power in a design with a similar number of measurements than in a design with more intervention measurements. In a design with a similar number of measurements both means will be rather stable estimates, while in a design with more intervention measurements the second case the estimate of the baseline mean will be relatively instable.

There is also an interesting interaction between the factor equal number of measurements and the ratio of the variation in scores in the phases. When there are less measurements in the baseline phase and there is more variation in scores in this phase, the power turned out to be

lower than any other combination of these two factors. This effect can be explained by the fact that a relatively small number of measurements and large variation will lead to unstable mean estimates which has a negative effect on the power.

The number of measurements clearly had an effect on the power of the randomization test. As marked before, more measurements lead to more stable means which is beneficial for the power. An interesting result is, though, that, other factors equal, the power increased in particular by moving from 15 to 30 observations. From 30 to 60 the power gain is small. This may be an important result from a practical point of view because collecting 60 measurements might be quite demanding.

The results of our study showed that the power of the randomization test was not different for correlated and uncorrelated baseline and intervention means across participants. So although the effect of autocorrelated data on the power was large, the effect of correlated baseline and intervention means across participants was absent. Since it often happens in practice that the means of the baseline and intervention scores are correlated across participants, it may be good to know that this correlation does not negatively affect the power.

In our crossed-factor simulation study we only simulated suddenly emerging intervention effects. That is, our intervention data were simulated from a normal distribution having a mean as large as the specified effect size. In practice the effect of the intervention may often be gradually instead of suddenly emerging. In a separate simulation study we differentiated between suddenly and gradually emerging intervention effects and the results showed that the effect of a gradually emerging effect on the power is very large. Based on these results we advise the researcher who is primary interested in the final effect of an intervention to exclude the measurements in which the effect is still emerging from the randomization test. This is of course only possible when the number of intervention measurements in which the effect has emerged is sufficiently large. One may check this and simulate the power of a specific design with a gradually emerging intervention effect using our online tool.

In our study, we focused on comparing individual means and the mean difference of the baseline and the intervention data formed distribution of the randomization test. We choose this test statistic because it is probably the most prevalent and well-known (e.g., [14, 41]). However, the randomization test is a distribution free test and does not require the test statistic to have a specific form. This characteristic offers a researcher to investigate other aspects of the data than the mean, such as the median and mode. Researchers might even test variation in scores in the baseline and intervention, ranges, and even regression lines or fluctuations over time may be interesting aspects of the data on which the baseline and the intervention phase can be compared. Obviously, the power results of this study can only be used for single case research where the mean difference is the statistic of interest but we think the next step is to investigate the power of randomization test for other statistics as well.

As elaborated on in the introduction, statistical testing was, and maybe still is, not an indisputable topic in the single case literature. Some researchers claim that data should not be aggregated at all, but shown graphically, direct and in absolute measures [58–59]. According to these researchers not only statistical testing should be in ban but also summarizing data in descriptive statistics leads to a loss in information and can therefore be misleading. We agree with these researchers in that recklessly grabbing some kind of mainstream statistic either for descriptive or inferential aims is bad practice. We also agree that–in general, not only in the context of single case designs, much more attention should be given to the visual representation of raw data before aggregating it to whatsoever statistic. We think, however, that it is not the descriptive or inferential statistics that are to be blamed, but the carelessness and ignorance in which these statistics are applied and interpreted. In our view descriptive statistics like several measures of effect size (see e.g. [8]), but also inferential statistics may add significant

incremental validity to the interpretation of data resulting from single case designs when these statistics are used correctly.

We believe that the randomization test, when used correctly, is a very flexible and complementary tool to value the statistical reliability of single case study outcome. We emphasize the correct use, because there are some pitfalls that are easily overlooked but may seriously hamper the interpretation of the statistical test. In order to correctly using a randomization test in a multiple baseline AB design it is required that one carefully specifies the range of possible start moments of the intervention *a priori* and then *randomly* draw the start moment of the intervention. This is important because in the randomization test it is assumed that each combination can actually occur in reality and that each combination of start moments has an equal probability to be drawn. When one of those aspects is not the case, a correct interpretation of the *p*-value cannot be guaranteed. In practice it may not be easy to define the range of possible start moments *a priori*. Take, for example, the clinical context where the baseline observations are collected when people are on the waiting list for a therapy. In this case it may be impractical, and even unethical to define a range of possible start moments beforehand and randomly draw one. Furthermore, in several contexts it may be inadequate or even impossible to randomly decide the start moment of the intervention. This is for example the case where an intervention has to start just after the baseline has reached stability. In these situation the randomization test discussed in this manuscript should not be used.

In this study we showed that, given that the above mentioned pitfalls are taken note of, the multiple baseline AB design might be powerful in many practical situations. As long as the observations within a participant are not too strongly correlated, neither the number of participants, the number of measurements and the expected effect size is too small, the randomization test has power to statistically evaluate a difference in baseline and intervention means. To conclude, we agree with many researchers on single case designs that statistical evidence should progress in tandem with the visual inspection analysis. To our opinion these two kinds of analyses are complementary rather than incompatible. For future research we would like to extend this study to other outcomes measures developed by visual inspection analysis.

## Supporting information

**S1 File. Some history of the randomization test.**
(DOCX)

**S2 File. Explanation of the Koehler and Levin (1998) randomization test.**
(DOCX)

**S3 File. R Code of the function used in the simulation study.**
(DOCX)

**S4 File. Autocorrelation and power in multiple baseline across subjects designs.**
(DOCX)

**S5 File. Correcting for gradually emerging patterns.**
(DOCX)

**S1 Table. Overview of possible (grey) and impossible (black) combinations of number of participants, number of start moments (k), number of measurements (t) and unique start moments or overlap in start moments.**
(DOCX)

**S2 Table. Mean and SD of the Type I error for all factor levels.**
(DOCX)

## Author Contributions

**Conceptualization:** Samantha Bouwmeester.

**Formal analysis:** Samantha Bouwmeester.

**Methodology:** Samantha Bouwmeester.

**Software:** Samantha Bouwmeester.

**Writing – original draft:** Samantha Bouwmeester, Joran Jongerling.

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
