## [Decision Letter · Decision Letter 0]

4 Nov 2019

PONE-D-19-24752

Power of a Randomization Test in a Single Case Multiple Baseline AB Design

PLOS ONE

Dear dr. Bouwmeester,

Thank you for submitting your manuscript to PLOS ONE. After careful consideration, we feel that it has merit but does not fully meet PLOS ONE’s publication criteria as it currently stands. Therefore, we invite you to submit a revised version of the manuscript that addresses the points raised during the review process.

We would appreciate receiving your revised manuscript by Dec 19 2019 11:59PM. To enhance the reproducibility of your results, we recommend that if applicable you deposit your laboratory protocols in protocols.io, where a protocol can be assigned its own identifier (DOI) such that it can be cited independently in the future. For instructions see: http://journals.plos.org/plosone/s/submission-guidelines#loc-laboratory-protocols

We look forward to receiving your revised manuscript.

Kind regards,

Mauro Gasparini, Ph.D.

Academic Editor

PLOS ONE

Journal Requirements:

Additional Editor Comments (if provided):

I agree with the referee report, please revise accordingly.

The request to shorten the paper is reasonable but it is not to be viewed as strict,

since PLOS ONE is an electronic journal.

Reviewers' comments:

Reviewer's Responses to Questions

**Comments to the Author**

1. Is the manuscript technically sound, and do the data support the conclusions?

Reviewer #1: Yes

2. Has the statistical analysis been performed appropriately and rigorously? 

Reviewer #1: Yes

3. Have the authors made all data underlying the findings in their manuscript fully available?

Reviewer #1: Yes

4. Is the manuscript presented in an intelligible fashion and written in standard English?

Reviewer #1: Yes

5. Review Comments to the Author

Reviewer #1: This paper deals with the issue of power evaluation for a randomization test in a single case multiple baseline AB design. It is well written, and clear and reasonably sophisticated. However, I have few concerns related to its publication in PLOS ONE.

Main concern:

As the authors state in the Discussion Section, a limitation of this study is that it only considers suddenly emerging intervention effects, that is, the introduction of the intervention leads to a sudden change of the score for the participants. But in practice often the effect of the intervention can be gradually developing. For example, in a clinical context, while it is possible that the effect of a medication can occur immediately after dosing, in many cases the effect of a medication can be negligible at first but then becomes apparent after certain time. Whether the conclusions in this paper can be generalized to the scenario of gradually effective intervention is unknown.

Also, based on the autoregressive function for the scores in the intervention on page 15, it seems that the authors assume a simple random walk model, that is, the score during the intervention is random and the best prediction for a score at time t is the score at the previous time. Intuitively this assumption is not very reasonable as it indicates that once the intervention is introduced, the effect is stable except for random fluctuations around its mean. But usually the dosing of a treatment can lead to an upward trend (if a positive change from baseline indicates improvement) or a downward trend (if a negative change from baseline indicates improvement). In my opinion this needs to be addressed through simulation studies.

Lesser concern

This manuscript is a bit long. The main part is 29 pages not including references, tables, figures and supporting information. Although PLOS ONE may not have a strict policy on page limit, it would be great if this manuscript can be more concise. For example, not exceeding 25 pages or can be even shorter (not including references, tables, figures and supporting information). Maybe the authors can shorten some paragraphs or move low priority content to supporting materials.

It is not very clear what the target audiences of this paper are. If the target audiences are statisticians, more statistical interpretations on the findings in the simulation studies are welcome. For example, the authors list a number of factors that impact the power, and their effects on power based on simulation study results. However, how to interpret the direction of these effects is not very clear. It seems that this paper has a mixed flavor of a statistical evaluation of the randomization test and a psychometric analysis.

Page 23: the authors conclude that "Having a large effect, Cohen’s d=1, six participants already result in an expected power of 0.8". This conclusion may be further validated through simulation studies varying other factors like the variance of the error or autoregressive function for the score.

Minor comments

Page 16: "the p value van not become" should be "the p value can not become".

6. PLOS authors have the option to publish the peer review history of their article (what does this mean?). If published, this will include your full peer review and any attached files.

Reviewer #1: No

---

## [Author Response · Author response to Decision Letter 0]

4 Dec 2019

PONE-D-19-24752

Power of a Randomization Test in a Single Case Multiple Baseline AB Design

PLOS ONE

Dear dr. Gasparini

We like to inform you that we have just submitted a revised version of our manuscript. Although the reviewer did not raise many points, we were glad with the comments because they were interesting and enabled us – hopefully - to improve the paper. In particular the first main concern of the reviewer, concerning the gradually emerging effects that often occur in a clinical context was an eye-opener for us and we decided that it would be worthwhile to add this as an additional factor in our simulation study. The results are quite striking we think, as the power of a randomization test turned out to drop hugely when the effect of an intervention is gradually emerging. We elaborated on this in the main text of the paper and added to the software the possibility that researcher can correct the randomization test for the number of measurements that the effect is gradually emerging. 

With respect to the second main concern we would like to mention the following. 

The purpose of adding simulations in which successive measures were (auto)correlated was to test the effect of violations of the assumption of interchangeable observations (i.e., independent successive observations)(within phases) made by the permutation test. To test this effect we didn’t generate data using a random walk model but based on an AR(1) model in which the AR-parameter varied between 0 and .5. We agree that trends are very likely in the type of data investigated. Change from pre- to post-intervention will likely be gradual and not instantaneous. This is why we also added gradual change to the simulations scenarios. However, we don’t focus on the effect of trends more extensively, because doing so would require studying the effect of different types of trends (i.e., linear, quadratic, etc.) and different variants of each type (e.g., steeper vs less steep linear change). As we are already studying the effect of a large number of different factors, we feel that adding these additional simulations would have a negative effect on the readability of the manuscript.

Instead, we plan on studying the effect of trends in more detail in a subsequent study.

With respect to the minor comments we like to write that we carefully reread the manuscript again and did an attempt to shorten it a bit. Unfortunately, because of the first main concern of the reviewer we had to add some text as well, so the net result is not really a shorter paper. Yet, we moved some parts to the Supporting Information. We’re happy to read that you mentioned that because PLOS ONE is a digital journal, the text length is of lesser concern. 

The reviewer also mentioned that it is not clear what the target audiences of this paper are. We emphasize that this paper is especially meant for clinical researchers who wants to use a randomization test for their multiple baseline design. The idea of the paper did actually stem from these researchers who raised the power question to us. However, to answer this power questions required quite some statistics and psychometrics that we had to discuss in the paper. In the new version of the manuscript we did an attempt to make even more clear that the study is meant for practical researchers. 

We agree with reviewer that it could be very helpful for practical researchers who want to use a randomization test for their multiple baseline design to freely vary all the factors in their particular design that – as the results of our study show - influence the power. We therefore add the possibility for researcher to do their own simulation study in our software. We think it is a real improvement that researcher are not restricted to the levels of the factors we incorporated in our simulation study but choose their own combination of factor levels to get an estimate of the power of their particular study. 

Finally, we corrected the typo at page 16 and some additional typo’s we found along the way. 

Kind regards,

Samantha Bouwmeester & Joran Jongerling

---

## [Decision Letter · Decision Letter 1]

14 Jan 2020

Power of a Randomization Test in a Single Case Multiple Baseline AB Design

PONE-D-19-24752R1

Dear Dr. Bouwmeester,

We are pleased to inform you that your manuscript has been judged scientifically suitable for publication and will be formally accepted for publication once it complies with all outstanding technical requirements.

With kind regards,

Mauro Gasparini, Ph.D.

Academic Editor

PLOS ONE

Additional Editor Comments (optional):

Reviewers' comments:

Reviewer's Responses to Questions

**Comments to the Author**

1. If the authors have adequately addressed your comments raised in a previous round of review and you feel that this manuscript is now acceptable for publication, you may indicate that here to bypass the “Comments to the Author” section, enter your conflict of interest statement in the “Confidential to Editor” section, and submit your "Accept" recommendation.

Reviewer #1: All comments have been addressed

2. Is the manuscript technically sound, and do the data support the conclusions?

Reviewer #1: Yes

3. Has the statistical analysis been performed appropriately and rigorously? 

Reviewer #1: Yes

4. Have the authors made all data underlying the findings in their manuscript fully available?

Reviewer #1: Yes

5. Is the manuscript presented in an intelligible fashion and written in standard English?

Reviewer #1: Yes

6. Review Comments to the Author

Reviewer #1: The authors addressed the reviewer’s comments well. I have two additional comments regarding the newly added contents on gradually emerging effect:

1. Page 13: the authors said that the effect-size of a gradually emerging effect is obviously smaller than of suddenly emerging effects. First, I think it should be ‘smaller than that of suddenly emerging effects’’, and secondly, this phrase itself is a bit confusing/misleading, because it is possible that the accumulative/overall effect size of a gradually emerging effect can be larger than the overall effect size of a suddenly emerging effect. Please clarify

2. Page 19: the authors said that the data was simulated with a suddenly emerging effect and gradually emerging effects during ¼, 1/3 and ¼ of the treatment measurements. Can the authors provide more details about how to implement this simulation procedure, either in the main manuscript or in the supporting materials?

A minor comment: Page 14: ‘The standard deviation within the baseline and intervention phase could be identical’ could be revised as ‘The standard deviation within the baseline phase and intervention phase could be identical’.

7. PLOS authors have the option to publish the peer review history of their article (what does this mean?). If published, this will include your full peer review and any attached files.

Reviewer #1: No

---

## [Editor Report · Acceptance letter]

29 Jan 2020

PONE-D-19-24752R1 

Power of a Randomization Test in a Single Case Multiple Baseline AB Design 

Dear Dr. Bouwmeester:

I am pleased to inform you that your manuscript has been deemed suitable for publication in PLOS ONE. Congratulations! Your manuscript is now with our production department. 

With kind regards,

on behalf of

Prof. Mauro Gasparini 

Academic Editor

PLOS ONE